# Unmet Supportive Care Needs after Non-Small Cell Lung Cancer Resection at a Tertiary Hospital in Seoul, South Korea

**DOI:** 10.3390/healthcare11142012

**Published:** 2023-07-12

**Authors:** Junhee Park, Wonyoung Jung, Genehee Lee, Danbee Kang, Young Mog Shim, Hong Kwan Kim, Ansuk Jeong, Juhee Cho, Dong Wook Shin

**Affiliations:** 1Department of Family Medicine & Supportive Care Center, Samsung Medical Center, Sungkyunkwan University School of Medicine, Seoul 06351, Republic of Korea; j-unicorn.park@samsung.com; 2Department of Family Medicine, Kangdong Sacred Heart Hospital, Hallym University, Seoul 05355, Republic of Korea; wyjung.md@kdh.or.kr; 3Department of Clinical Research Design and Evaluation, Samsung Advanced Institute for Health Sciences and Technology (SAIHST), Sungkyunkwan University, Seoul 06351, Republic of Korea; genehee.lee@samsung.com (G.L.); dbee.kang@samsung.com (D.K.); jh1448.cho@samsung.com (J.C.); 4Patient-Centered Outcomes Research Institute, Samsung Medical Center, Seoul 06351, Republic of Korea; 5Department of Thoracic and Cardiovascular Surgery, Samsung Medical Center, Seoul 06351, Republic of Korea; youngmog.shim@samsung.com (Y.M.S.); hkts.kim@saumsung.com (H.K.K.); 6Department of Psychology, DePaul University, Chicago, IL 60604, USA; ajeong@depaul.edu; 7Cancer Education Center, Samsung Medical Center, Seoul 06351, Republic of Korea; 8Department of Digital Health, Samsung Advanced Institute for Health Science & Technology (SAIHST), Sungkyunkwan University School of Medicine, Seoul 06355, Republic of Korea

**Keywords:** unmet need, fear of cancer recurrence, quality of life, lung cancer, cancer survivor

## Abstract

The aim of this study is to identify the prevalence and predictors of unmet needs of non-small cell lung cancer (NSCLC) patients undergoing surgical resection in Seoul, South Korea. A total of 949 patients who completed survey questionnaires that included the Cancer Survivors’ Unmet Needs Korean version (CaSUN-K), fear of cancer recurrence (FCR) inventory-short form, and European Organization for Research and Treatment of Cancer Quality of Life Core Questionnaire (EORTC QLQ-C30) were recruited from January to October 2020. Multivariable logistic regression was used to determine the potential correlation of significant unmet needs, defined as any moderate or strong need, for each domain of CaSUN-K. Of the 949 participants, the mean age was 63.4 ± 8.8 years old, and 529 (55.7%) were male. Overall, 91.8% of participants reported one or more unmet need. The highest domains of moderate-to-strong unmet needs were existential survivorship (59.1%), comprehensive cancer care (51.2%), and information (49.7%). High FCR and poor emotional function were associated with moderate-to-strong unmet needs across all domains of CaSUN-K. NSCLC survivors with a recent diagnosis had more frequent disease-related unmet needs. Interventions to reduce the unmet needs of NSCLC patients should focus on relieving FCR and improving emotional functioning. Furthermore, emphasis should be placed on decreasing disease-related needs, particularly for early survivors of lung cancer during the re-entry phase.

## 1. Introduction

Lung cancer is the second most commonly diagnosed cancer (11.4% of total cases) and the leading cause of cancer death (18.0% of total cancer deaths) worldwide [1]. Due to improvements in early cancer detection [2,3] and advances in intensive multidisciplinary treatments [4] that have improved survival rates, the number of cancer survivors is expected to increase. Most survivors receive treatment such as surgery, radiation therapy, and/or chemotherapy and experience a higher burden of comorbidity [5]. Cancer treatment does not guarantee a full recovery; thus, many survivors experience less than adequate overall and health-related quality of life (QoL) with potential long-term decline [5,6].

Cancer-related outcomes can have a significant effect on the psychological experiences of cancer survivors such as distress, anxiety, depression, fear of cancer recurrence (FCR), social support, relationship issues, and QoL [7]. As awareness of the psychosocial effect of cancer increases, attempts have been made to assess individual needs more broadly to improve supportive care services [8,9,10]. Reportedly, lung cancer patients have a greater number of unmet needs compared with patients with other types of cancer [11] and constitute a unique population of cancer patients who likely have greater disease-specific supportive care needs such as information on managing symptoms such as dyspnea and cough [12].

In a recent systematic review [13] in which six studies of unmet supportive care needs in lung cancer patients were examined using a standardized tool, unmet needs were associated with poorer QoL [13]. Among the studies, the unmet needs based on the Cancer Survivors’ Unmet Needs (CaSUN) scale [13] were investigated in only one study, and the moderate-to-strong unmet needs of various cancer survivors, including lung cancer, in Asian countries were reported [14]. However, the authors only reported results collected from the unmet needs survey and did not perform any further analysis such as exploring correlates of unmet needs in each domain. Based on the identification of unique needs using domain-specific analysis, the results of the present study could aid in understanding the unmet needs and address suitable supportive care for lung cancer patients.

In Korea, Yun et al. previously attempted to determine several significant correlates of unmet needs in Korean lung cancer survivors including socioeconomic burden (female, employment, and fewer family members) and medical burden (chemotherapy and long disease duration after cancer diagnosis) [15]. However, in that study, a validated instrument for assessing unmet needs was not used, and psychological factors known to be associated with unmet needs, such as QoL, were not evaluated [13].

Therefore, in the present study, the prevalence of unmet needs of lung cancer survivors was evaluated and the correlates, including demographic, disease-related, and psychological factors, associated with unmet needs in different domains were explored. The results provide detailed information regarding the main correlates of unmet needs and show how NSCLC survivors’ unmet needs can be best addressed.

## 2. Materials and Methods

### 2.1. Study Design, Setting, and Population

This cross-sectional study recruited 1220 patients using convenience sampling from the outpatient clinics of thoracic surgery and long-term survivorship at the Samsung Comprehensive Cancer Center in South Korea from January to October 2020. Among them, 1014 (83.1%) provided informed consent to participate the study. Individuals who were diagnosed with primary non-small cell lung cancer (NSCLC) and received curative pulmonary resection with or without neo-adjuvant treatment were enrolled in this study. Cancer patients <18 years of age or not able to communicate in Korean were excluded. Patients who did not complete treatment within at least 1 month before the study participation, experienced recurrence/metastasis of cancer, or had secondary cancer were also excluded from the study. Finally, a total of 949 patients was included in the final analysis.

### 2.2. Data Collection

The medical records of participants were collected, and cancer-related information was comprehensively reviewed such as the time since the end of active treatments, the stage of cancer, and all types of cancer treatments received. A trained research assistant conducted a face-to-face interview to complete a questionnaire that included the CaSUN Korean version (CaSUN-K), the Korean version of the FCR (K-FCRI-SF) Inventory-Short Form, and the Korean version of European Organization for Research and Treatment of Cancer Quality of Life Core Questionnaire (EORTC QLQ-C30). Sociodemographic characteristics items such as age, sex, marital status, educational level, monthly family income, and work status were also included in the questionnaire.

### 2.3. Measurement of Unmet Need

The original CaSUN contains 35 items used to evaluate the supportive care needs of cancer survivors across five dimensions: information (3 items), comprehensive cancer care (6 items), existential survivorship (14 items), quality of life (2 items), and relationship (3 items) [16]. For each item, participants answer whether or not they had a particular need and, if they did, whether or not that need was met or unmet. If an unmet need was reported, the participant then rated the intensity of the need as weak, moderate, or strong. In the present study, moderate-to-strong ratings of unmet needs were mainly discussed based on several previous studies [17,18,19].

In a previous study, CaSUN-K was shown to be a valid measurement tool for unmet needs in Korean NSCLC survivors [20]. In the CaSUN-K, the financial dimension was newly added. In addition, 7 items that were originally not grouped into any dimension were distributed; 4 of them were classified into the financial dimension in CaSUN-K [20] (Figure 1).

### 2.4. Other Measures

The K-FCRI-SF has been validated and widely used to measure FCR severity in cancer survivors in Korea [21,22]. The 9-item K-FCRI-SF is rated on a 5-point Likert scale from 0 to 4, and total scores range from 0 to 36. The higher the score, the higher the FCR. We defined higher FCR as a K-FCRI-SF score ≥13 points based on a previous study [23].

The EORTC QLQ-C30 has been translated into Korean, validated, and widely used to measure the QoL of cancer survivors in Korea [24]. Each item was rated based on a symptom score ranging from 0 to 100. The higher the score, the better the functional level. A cut-off score of 66.7 was used to define poor function based on previous studies [25,26].

### 2.5. Statistical Analysis

Descriptive statistics were used to describe the domains of unmet needs as well as all other variables. Univariable and multivariable logistic regression analyses were used to analyze the potential correlates of each domain of the unmet need. Known confounding factors of age, sex, smoking, educational level, marital status, working status, family monthly income, comorbidity, neo-adjuvant treatment, pathologic staging, adjuvant treatment, time since the end of active treatment, FCR, and functioning collected from EORTC QLQ-C30 were included in the multivariable logistic models using stepwise backward selection. Stratified analyses were performed to determine the differences between lengths of survival. All statistical analyses were conducted using STATA/MP 14.0 (Stata Corp., College Station, TX, USA).

## 3. Results

### 3.1. General Characteristics of Study Participants

Among the 949 participants, the mean age was 63.4 ± 8.8 years old, and 529 (55.7%) were male. Among them, 48.9% were never smokers and 41.9% were employed. The median time since the end of active treatment was 18 months. A high FCR was found in 530 (55.8%) participants. In terms of the EORTC QLQ-C30 function scale, poor physical function (17.7%), poor role function (40.5%), poor emotional function (30.9%), poor cognitive function (25.7%), and poor social function (35.0%) were observed (Table 1).

### 3.2. Prevalence of Unmet Needs

The CaSUN-K identifies ranges and domains of different needs, allowing comparisons across the lung cancer survivor population. The prevalence of at least one moderate-to-strong unmet need in each domain of CaSUN-K was 59.1% for existential survivorship, 51.2% for comprehensive cancer care, 49.7% for information, 38.9% for QoL, 31.0% for relationship, and 22.6% for financial issue (Figure 1). In order of frequency, the top five moderate-to-strong unmet need items were “concerns about cancer recurrence” (51.5%) in the existential survivorship domain, “communication among doctors” (49.8%), “manage health with team” (48.4%), and “local healthcare services” (44.5%) in the comprehensive cancer care domain, and “understandable information” (44.3%) in the information domain. The higher ranked met needs were “best medical care” (16.2%) and “manage health with team” (15.1%) in the comprehensive cancer care domain; “reduce stress in my life” (15.0%) and “survivor expectations” (14.8%) in the existential survivorship domain; and “up-to-date information” (14.2%) in the information domain (Table 2).

In Table 3, information needs were associated with high FCR, poor emotional function, and poor cognitive function. Comprehensive cancer care needs were associated with male gender, shorter time since the end of active treatment, high FCR, poor emotional function, and poor social function. Existential survivorship needs were associated with shorter time since the end of active treatment, high FCR, poor physical function, poor emotional function, and poor social function. QoL needs were associated with shorter time since the end of active treatment, high FCR, poor emotional function, and poor social function. Relationship needs were associated with high FCR and poor emotional and poor social function. Financial needs were associated with young age, high FCR, poor role function, and poor emotional function. In a sensitivity analysis by length of survival dichotomized as <18 and ≥18 months (Appendix A), high FCR was associated with moderate-to-strong unmet needs across all domains. Poor social function was significant in NSCLC patients with shorter survival duration (<18 months), while poor emotional function was significant in those with longer survival duration (≥18 months).

## 4. Discussion

In the present cross-sectional study, the prevalence and correlates of unmet needs in NSCLC survivors were investigated. 91.8% of participants reported one or more unmet need. The highest domains of unmet needs were existential survivorship (59.1%), comprehensive cancer care (51.2%), and information (49.7%), in agreement with the Australian population of the CaSUN [16].

Lung cancer patients in this study in Korea were likely to have disease-specific supportive care needs, similar to a previous study [12]. In detail, the leading moderate-to-strong unmet needs were associated with disease cure or health service such as “concerns about cancer recurrence” in the existential survivorship domain; “communication among doctors”, “manage health with team”, and “local healthcare services” in the comprehensive cancer care domain; and “understandable information” in the information domain. These results were consistent in the order of frequency with a previous study of patients across the Asia–Pacific region [14], in which the top five moderate-to-strong unmet needs in Korea were “concerns about cancer recurrence” in the existential survivorship domain and “local healthcare services”, “best medical care”, “manage health with team”, and “communication among doctors” in the comprehensive cancer care domain.

Notably, the “best medical care” was the highest met need. This difference could have resulted from the quality of the investigating institute, which was a university-affiliated hospital with a separate cancer center providing cancer care for approximately 15% of Korean lung cancer patients [27].

### 4.1. Correlates of Unmet Needs

High FCR and poor emotional function were associated with moderate-to-strong unmet needs across all domains of CaSUN-K. In previous studies, “managing concerns about cancer recurrence” was the top unmet need in lung cancer survivors [12,14,28]. Most cancer patients fear that illness will progress [29,30]. Furthermore, high FCR remained significant in sensitivity analysis by length of survival. Survivors who have a physical condition or have received medical treatment that interfered with their social life had an increased risk of unmet needs across all domains in those with a shorter duration of survival. However, survivors gradually adapt over time; those with poor emotional function such as depressed are at risk of having moderate-to-strong unmet needs across all domains. The emotional function in EORTC QLQ-C30 consists of questions regarding patient tension, worry, irritability, and/or depression. Anxiety and depression were the most significant factors in unmet supportive care needs in a study from Taiwan [31]. A higher level of unmet needs was associated with lower perceived QoL [16] in various cancer survivors [13,32,33,34,35]. Therefore, interventions to relieve the FCR and improve emotional function should be performed to reduce unmet needs in NSCLC survivors.

Information needs were associated with poor cognitive function. Presumably, a greater amount of information is required for comprehension in patients with decreased cognitive function. In a previous study in Iran [36], the reasons for insufficient fulfillment of information needs in subjects with poor cognitive abilities were outlined. A concern expressed by patients was that their lack of understanding regarding health hindered their ability to effectively communicate with medical personnel during the treatment process, resulting in a potential misinterpretation of the medical information provided by the experts [36]. Therefore, information needs of NSCLC patients with cognitive impairments should be taken into consideration.

Comprehensive cancer care needs were associated with male gender, shorter duration after treatment, and poor social function. The domain of comprehensive cancer care consists of items associated with disease cure or health service such as “best medical care”, “local healthcare services”, “manage health with team”, and “communication among doctors” that were closely linked to the health system of supportive care needs (SCNS) [8,9]. Male patients had greater expectations for sexual information and psychosocial support [37] as well as for health system-related needs than female patients. In addition, an individual with a recent diagnosis reportedly had more unmet needs associated with disease [14,38]. Thus, healthcare providers should focus on patients, particularly early survivors of NSCLC, during the re-entry phase. Previously, the social function in EORTC-QLQ-C30 was associated with domains of comprehensive cancer care (beta coefficient, −0.27), existential survivorship (−0.39), QoL (−0.32), and relationship (−0.28) in CaSUN-K [20].

Existential survivorship needs were predicted based on shorter disease duration after treatment, poor physical function, and poor social function. The highest unmet needs were found in the existential survivorship domain with the “concerns about cancer recurrence” item accounting for more than half of the unmet needs. In the present study, physical function and social function were shown to play an important role in addressing the unmet needs in existential survivorship. This highlights the need for healthcare professionals to provide comprehensive care to NSCLC patients with poor function in physical and social areas.

QoL needs were associated with a shorter duration after treatment and poor social function. The QoL domain included items such as “manage side effects” and “changes to QoL”. The results of the present study add to the present knowledge that early survivors of NSCLC appear to be more focused on side effects to improve their QoL needs.

Relationship needs were associated with poor social function. The relationship domain included items relating to social issues such as “support partner or family”, “impact on my relationship”, “changes to partner’s life”, and sexual issues such as “problems with sex life”. Individuals who struggle with social interactions may have unfulfilled social-related needs.

Financial needs were associated with young age and poor role function. This is consistent with findings in previous studies [20,39] that younger patients are expected to have a greater need for rehabilitation and to experience multiple challenges such as losing a full-time job or not fulfilling their family responsibilities [20,40,41]. Similarly, patients with poor role function who want to contribute to their family but cannot do so may be at risk of unmet needs in the financial issue domain.

### 4.2. Clinical Implications

The results of the present study contribute to the existing knowledge by highlighting that unmet supportive care needs in lung cancer patients are associated with poorer psychological factors such as QoL. FCR and poor emotional function were associated with unmet needs across all domains. Therefore, interventions of supportive care for NSCLC patients such as cognitive behavioral therapy or patient education should be focused on relieving FCR and improving emotional and social functioning. Furthermore, caregivers should prioritize supportive care to early survivors of NSCLC during the re-entry phase to reduce unmet supportive care needs.

### 4.3. Limitations

The present study had several limitations. First, the study cohort might not represent the general NSCLC survivor population because it included only patients from one tertiary hospital recruited using convenience sampling. In addition, patients who had previously experienced second cancer or cancer recurrence/metastasis were excluded. Therefore, a selection bias may limit the generalization of these findings to other NSCLC survivors. Second, the study was conducted based on self-reported measures, so there could be social desirability and recall biases. Third, there is no consensus on the standard cut-off point of the EORTC-QLQ-C30 with cancer patients. A cut-off point <66.7 was used in the present study based on the population distribution of scores. In addition, this was an observational study, and the cross-sectional settings restrict the conclusion of any causal relationship between unmet needs and psychological traits such as FCR and poor functioning. Reciprocal relationships between unmet needs and QoL are also possible. Cohort studies are needed to elucidate such complex relationships. Finally, there was a potential for unadjusted confounding factors.

## 5. Conclusions

In the present study, the correlates of unmet needs were investigated using validated instruments including psychological variables with a large number of NSCLC patients in Seoul, South Korea. The findings indicated that higher FCR and poor emotional and social functioning were correlates of unmet needs in most domains. In addition, emphasis should be placed on decreasing disease-related needs, particularly of early lung cancer survivors. Therefore, interventions should aim to alleviate the FCR and enhance the emotional and social well-being of lung cancer patients by addressing their unmet needs.

## Figures and Tables

**Figure 1 healthcare-11-02012-f001:**
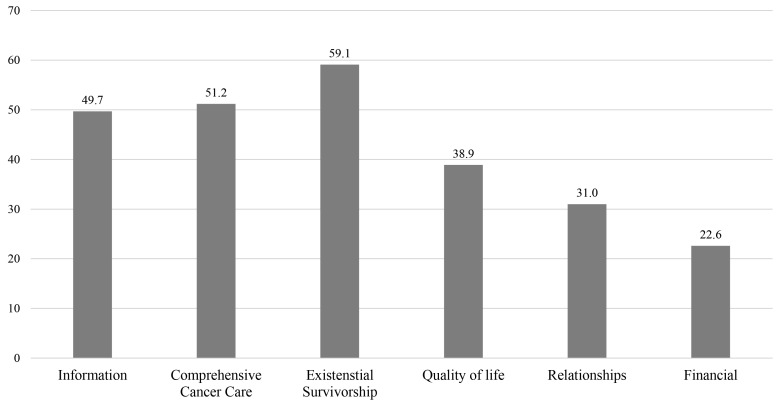
Moderate-to-strong unmet needs of CaSUN-K’s dimensions.

**Table 1 healthcare-11-02012-t001:** Baseline characteristics of the total study population.

Characteristics	Total Study Population (*n* = 949)
	Number (%)
Sociodemographic characteristics	
Mean age, years *	63.4 ± 8.8
<65	510 (53.7)
≥65	439 (46.3)
Sex	
Male	529 (55.7)
Female	420 (44.3)
Smoking status	
Never smoker	464 (48.9)
Ex- or current smoker	485 (51.1)
Education level	
≥University	353 (37.2)
High school	328 (34.6)
≤Middle school	267 (28.1)
Unknown	1 (0.1)
Marital status	
Married	820 (86.4)
Working status	
Unemployed	551 (58.1)
Employed	398 (41.9)
Household monthly income	
<USD$3000	335 (35.3)
≥USD$3000	536 (56.5)
Unknown	78 (8.2)
Disease-related characteristics	
Comorbidity	
No	239 (25.2)
Yes	710 (74.8)
Pathologic stage	
0	12 (1.3)
I	697 (73.4)
II	140 (14.8)
III **	100 (10.5)
Time since the end of active treatment	
<18 months	461 (48.6)
≥18 months	488 (51.4)
Psychological characteristics	
Fear of cancer recurrence (FCR)	
Low FCR (<13)	419 (44.2)
High FCR (≥13)	530 (55.8)
Function domain of EORTC-QoL-C30	
Poor Physical Function	168 (17.7)
Poor Role Function	384 (40.5)
Poor Emotional Function	293 (30.9)
Poor Cognitive Function	244 (25.7)
Poor Social Function	332 (35.0)

Abbreviation: FCR = Fear of cancer recurrence; EORTC-QoL-C30 = European Organization for Research and Treatment of Cancer Quality of Life core questionnaire. * Data are expressed as mean ± standard deviation. ** Subjects with metastasis are excluded.

**Table 2 healthcare-11-02012-t002:** Severity according to the Korean version of the Cancer Survivor’s Unmet Needs (CaSUN-K).

CaSUN-K’s Survey Items	Question Number	No Need	Met Need	Unmet Need, Weak	Unmet Need, Moderate	Unmet Need, Strong	Unmet Need Moderate-to-Strong
		Number (%)	Number (%)	Number (%)	Number (%)	Number (%)	Number (%)
Information							
Up-to-date information	1	178 (18.8)	135 (14.2)	239 (25.2)	178 (18.8)	219 (23.1)	397 (41.8)
Information for others	2	221 (23.3)	108 (11.4)	256 (27.0)	177 (18.7)	187(19.7)	364 (38.4)
Understandable information	3	176 (18.5)	98 (10.3)	255 (26.9)	188 (19.8)	232 (24.4)	420 (44.3)
Comprehensive cancer care							
Best medical care	4	238 (25.1)	154 (16.2)	175 (18.4)	172 (18.1)	210 (22.1)	382 (40.3)
Local healthcare services	5	228 (24.0)	124 (13.1)	175 (18.4)	192 (20.2)	230 (24.2)	422 (44.5)
Manage health with team	6	144 (15.2)	143 (15.1)	203 (21.4)	193 (20.3)	266 (28.0)	459 (48.4)
Communication among doctors	7	145 (15.3)	131 (13.8)	200 (21.1)	200 (21.1)	273 (28.8)	473 (49.8)
Complaints are addressed	8	236 (24.9)	127 (13.4)	176 (18.5)	185 (19.5)	225 (23.7)	410 (43.2)
Complimentary therapy *	9	405 (42.7)	80 (8.4)	199 (21.0)	134 (14.1)	131 (13.8)	265 (27.9)
Accessible hospital parking	18	557 (58.7)	95 (10.0)	95 (10.0)	66 (7.0)	136 (14.3)	202 (21.3)
Existential survivorship							
Reduce stress in my life	10	274 (28.9)	142 (15.0)	262 (27.6)	150 (15.8)	121 (12.8)	271 (28.6)
Concerns about cancer coming back	19	155 (16.3)	71 (7.5)	234 (24.7)	159 (16.8)	330 (34.8)	489 (51.5)
Emotional support for me	20	352 (37.1)	102 (10.7)	218 (23.0)	166 (17.5)	111 (11.7)	277 (29.2)
New relationships	23	470 (49.5)	107 (11.3)	184 (19.4)	102 (10.7)	86 (9.1)	188 (19.8)
Talk to others	24	355 (37.4)	110 (11.6)	230 (24.2)	138 (14.5)	116 (12.2)	254 (26.8)
Handle social/work situations	25	462 (48.7)	100 (10.5)	196 (20.7)	109 (11.5)	82 (8.6)	191 (20.1)
Changes to my body	26	371 (39.1)	113 (11.9)	234 (24.7)	142 (15.0)	89 (9.4)	231 (24.3)
Ongoing case manager *	28	490 (51.6)	68 (7.2)	167 (17.6)	110 (11.6)	114 (12.0)	224 (23.6)
Move on with my life	29	424 (44.7)	124 (13.1)	216 (22.8)	105 (11.1)	80 (8.4)	185 (19.5)
Changes to beliefs	30	422 (44.5)	111 (11.7)	228 (24.0)	114 (12.0)	74 (7.8)	188 (19.8)
Acknowledging the impact	31	518 (54.6)	121 (12.8)	165 (17.4)	85 (9.0)	60 (6.3)	145 (15.3)
Survivor expectations	32	415 (43.7)	140 (14.8)	209 (22.0)	101 (10.6)	84 (8.9)	185 (19.5)
Decisions about my life	33	467 (49.2)	110 (11.6)	192 (20.2)	103 (10.9)	77 (8.1)	180 (19.0)
Spiritual beliefs	34	546 (57.5)	136 (14.3)	134 (14.1)	82 (8.6)	51 (5.4)	133 (14.0)
Make my life count	35	476 (50.2)	117 (12.3)	189 (19.9)	96 (10.1)	71 (7.5)	167 (17.6)
Quality of life							
Manage side effects	11	352 (37.1)	83 (8.7)	201 (21.2)	136 (14.3)	177 (18.7)	313 (33.0)
Changes to quality of life	12	300 (31.6)	112 (11.8)	235 (24.8)	143 (15.1)	159 (16.8)	302 (31.8)
Relationships							
Support partner/family	21	410 (43.2)	100 (10.5)	203 (21.4)	133 (14.0)	103 (10.9)	236 (24.9)
Impact on my relationship	22	450 (47.4)	92 (9.7)	188 (19.8)	122 (12.9)	97 (10.2)	219 (23.1)
Changes to partner’s life *	13	900 (94.8)	23 (2.4)	11 (1.2)	6 (0.6)	9 (0.9)	15 (1.6)
Problems with sex life	27	707 (74.5)	66 (7.0)	109 (11.5)	44 (4.6)	23 (2.4)	67 (7.1)
Financial issues (new dimension)							
Impact on my working life *	14	729 (76.8)	59 (6.2)	78 (8.2)	37 (3.9)	46 (4.8)	83 (8.7)
Financial support *	15	502 (52.9)	77 (8.1)	141 (14.9)	104 (11.0)	125 (13.2)	229 (24.1)
Life/travel insurance *	16	584 (61.5)	73 (7.7)	139 (14.6)	71 (7.5)	82 (8.6)	153 (16.1)
Legal services *	17	600 (63.2)	53 (5.6)	118 (12.4)	92 (9.7)	86 (9.1)	178 (18.8)

* Seven items, originally not grouped into any dimension, were distributed into the 1 new or an existing dimension.

**Table 3 healthcare-11-02012-t003:** Predictors of moderate-to-strong level of unmet needs of CaSUN-K in each domain by uni- and multivariable logistic regression analyses.

	Information		Comprehensive Cancer Care		Existential Survivorship		Quality of Life		Relationship		Financial Issues	
	OR (95% CI)	aOR * (95% CI)	OR (95% CI)	aOR * (95% CI)	OR (95% CI)	aOR * (95% CI)	OR (95% CI)	aOR * (95% CI)	OR (95% CI)	aOR * (95% CI)	OR (95% CI)	aOR * (95% CI)
Sociodemographic characteristics												
Age												
≥65 years	1.00		1.00		1.00		1.00		1.00		1.00	
<65 years	1.09 (0.85, 1.41)		1.09 (0.84, 1.40)		1.20 (0.93, 1.56)		1.19 (0.91, 1.55)		1.17 (0.89, 1.55)		1.48 (1.09, 2.02)	1.54 (1.09, 2.18)
Sex												
Female	1.00		1.00		1.00		1.00		1.00		1.00	
Male	0.88 (0.67, 1.13)		1.11 (0.86, 1.43)	1.37 (1.01, 1.85)	0.77 (0.59, 1.00)		0.83 (0.63, 1.07)		1.00 (0.76, 1.32)		0.92 (0.68, 1.25)	
Marital status												
Non-married	1.00		1.00		1.00		1.00		1.00		1.00	
Married	0.81 (0.56, 1.18)		1.24 (0.86, 1.80)		0.61 (0.41, 0.91)		0.75 (0.51, 1.09)		1.04 (0.70, 1.56)		0.40 (0.27, 0.59)	
Education level												
≥University	1.00		1.00		1.00		1.00		1.00		1.00	
High school	1.00 (0.74, 1.35)		1.10 (0.82, 1.49)		1.08 (0.80, 1.47)		1.01 (0.74, 1.38)		0.92 (0.67, 1.28)		1.02 (0.71, 1.45)	
≤Middle school	1.09 (0.79, 1.50)		1.05 (0.77, 1.45)		1.04 (0.75, 1.44)		1.06 (0.76, 1.46)		0.86 (0.61, 1.21)		0.86 (0.59, 1.27)	
Employment status												
Employed	1.00		1.00		1.00		1.00		1.00		1.00	
Unemployed	1.07 (0.83, 1.39)		0.90 (0.69, 1.16)		1.29 (0.99, 1.68)		1.09 (0.84, 1.42)		1.16 (0.88, 1.54)		0.97 (0.71, 1.32)	
Household monthly income												
<USD$3000	1.00		1.00		1.00		1.00		1.00		1.00	
≥USD$3000	0.91 (0.69, 1.19)		1.03 (0.78, 1.35)		0.79 (0.60, 1.05)		0.92 (0.70, 1.22)		0.90 (0.67, 1.20)		0.76 (0.55, 1.05)	
Disease-related characteristics												
Comorbidity												
No	1.00		1.00		1.00		1.00		1.00		1.00	
Yes	1.12 (0.83, 1.50)		1.21 (0.90, 1.62)		1.13 (0.84, 1.52)		1.27 (0.93, 1.72)		1.06 (0.77, 1.45)		0.80 (0.57, 1.13)	
Pathologic stage												
0–I	1.00		1.00		1.00		1.00		1.00		1.00	
II	1.25 (0.87, 1.79)		1.09 (0.76, 1.56)		1.23 (0.85, 1.79)		1.37 (0.95, 1.97)		1.47 (1.01, 2.14)		1.32 (0.87, 2.01)	
III	1.37 (0.80, 1.07)		1.38 (0.90, 2.10)		1.67 (1.07, 2.62)		1.92 (1.26, 2.92)		1.19 (0.76, 1.86)		1.72 (1.08, 2.72)	
Lapse after active treatment												
≥18 months	1.00		1.00		1.00		1.00		1.00		1.00	
<18 months	1.49 (1.16, 1.93)		1.58 (1.22, 2.04)	1.50 (1.11, 2.02)	1.65 (1.27, 2.14)	1.55 (1.12, 2.14)	1.70 (1.31, 2.21)	1.59 (1.17, 2.15)	1.43 (1.09, 1.89)		1.55 (1.14, 2.11)	
Psychological characteristics												
Fear of Cancer Recurrence												
Low FCR (<13)	1.00		1.00		1.00		1.00		1.00		1.00	
High FCR (≥13)	4.64 (3.52, 6.12)	4.13 (3.02, 5.64)	4.29 (3.26, 5.64)	3.87 (2.83, 5.30)	7.05 (5.28, 9.41)	5.99 (4.32, 8.30)	4.97 (3.69, 6.68)	3.90 (2.79, 5.46)	5.44 (3.90, 7.57)	4.39 (3.02, 6.39)	3.89 (2.72, 5.57)	2.77 (1.85, 4.15)
EORTC-QoL-C30												
Poor Physical Function	2.27 (1.60, 3.21)		1.92 (1.36, 2.71)		3.13 (2.10, 4.67)	1.64 (1.01, 2.65)	2.14 (1.53, 3.00)		1.78 (1.26, 2.50)		1.71 (1.18, 2.48)	
Poor Role Function	2.22 (1.71, 2.90)		2.05 (1.57, 2.67)		2.19 (1.66, 2.88)		2.19 (1.68, 2.86)		2.30 (1.74, 3.04)		2.45 (1.80, 3.35)	1.56 (1.09, 2.24)
Poor Emotional Function	3.47 (2.58, 4.66)	1.93 (1.35, 2.77)	3.11 (2.32, 4.17)	1.70 (1.18, 2.44)	3.99 (2.89, 5.52)	1.65 (1.09, 2.49)	3.53 (2.65, 4.71)	1.86 (1.30, 2.65)	3.65 (2.72, 4.89)	1.87 (1.31, 2.68)	3.73 (2.72, 5.13)	2.31 (1.60, 3.35)
Poor Cognitive Function	2.64 (1.95, 3.59)	1.65 (1.18, 2.31)	2.38 (1.75, 3.23)		2.67 (1.92, 3.70)		2.17 (1.62, 2.92)		2.32 (1.72, 3.15)		2.39 (1.72, 3.30)	
Poor Social Function	2.73 (2.07, 3.61)		2.87 (2.17, 3.80)	1.89 (1.34, 2.66)	4.18 (3.07, 5.70)	2.34 (1.59, 3.44)	3.27 (2.48, 4.33)	2.00 (1.42, 2.80)	3.01 (2.26, 4.01)	1.79 (1.26, 2.53)	2.57 (1.88, 3.51)	

Abbreviation: CaSUN-K = Korean version of the Cancer Survivors’ Unmet Needs, EORTC-QoL-C30 = European organization for research and treatment of cancer quality of life core questionnaire, OR = odds ratio, aOR = adjusted odds ratio, CI = confidence interval. * Data are expressed based on multivariable logistic models by stepwise backward selection considering significance at *p* < 0.054.

## Data Availability

The data that support the findings of this study are available from the corresponding author upon request.

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
