# Peer review of "Unmet Supportive Care Needs after Non-Small Cell Lung Cancer Resection at a Tertiary Hospital in Seoul, South Korea"

_healthcare, 2023, doi:10.3390/healthcare11142012_

Round 1

Author Response

Response to comments from the Reviewer 1

Comment #1. The authors have explored the unmet needs in cancer care among lung cancer patients in a single hospital setting in Korea and added to the existing evidence. A few more details in the manuscript would improve this work.

Response: Thank you for the insightful comment. We changed the title to “Unmet supportive care needs after non-small cell lung cancer resection at a tertiary hospital in Seoul, South Korea.”

[Title] (page 1, line 2)

Unmet supportive care needs after non-small cell lung cancer resection at a tertiary hospital in Seoul, South Korea

Comment #2. Please provide some examples of disease-specific supportive care needs.

Response: We appreciate your thoughtful comments and suggestions. The previous study that we cited reported that the most desired information needs among lung cancer patients were those for information on managing symptoms of fatigue (78%), shortness of breath (77%), and cough (63%) (Giuliani et al., 2016). A recent systematic review of unmet supportive care needs in lung cancer patients demonstrated that the symptom burden experienced by lung cancer patients was greater than those of other types of cancer (Cochrane et al, 2022; Molassiotis et al, 2017). Symptoms associated with the range of treatments and the disease itself include pain, dyspnea, fatigue, and anorexia. These symptoms are associated with impaired functioning and have a negative impact on quality of life. Individuals who receive a lung cancer diagnosis experience high rates of anxiety and depression and are at a greater risk for suicide compared with other individual with other types of cancer (Anguiano et al., 2012). This may, in part, be related to the side-effects of aggressive treatment, poor prognosis, and the stigma associated with smoking behaviors. A previous study reported that lung cancer patients were satisfied with the cancer-related information they received, such as prognosis and supportive care for complication such as pain and fatigue (Yun et al., 2013). We added examples of disease-specific supportive care needs in the manuscript.

[References]

Giuliani ME, Miline RA, Puts M, Sampson LR, Kwan JY, Le LW, et al. The prevalence an dnature of supportive care needs in lung cacner patients. Curr Oncol. 2016;23:258-65.

Anguiano L, Mayer DK, Piven ML, Rosenstein D. A literature review of suicide in cancer patients. Cancer Nurs. 2012;35(4): E14-E26.

Cochrane A, Woods S, Dunne S, Gallagher P. Unmet supportive care needs associated with quality of life for people with lung cancer: A systematic review of the evidence 2007-2020. Eur J Cancer Care (Engl). 2022;31:e13525.

Yun YH, Shon EJ, Yang AJ, Kim SH, Kim YA, Chang YJ, et al. Needs regarding care and factors associated with unmet needs in disease-free survivors of surgically treated lung cancer. Ann Oncol. 2013;24:1552-9.

[Introduction] (page 2, paragraph 2, line 54)

Reportedly, lung cancer patients have a greater number of unmet needs compared with patients with other types of cancer and constitute a unique population of cancer patients who likely have greater disease-specific supportive care needs such as information on managing symptoms like dyspnea and cough

Comment #3. Why were patients with other subtypes not considered? Was it because of differences in survival? NSCLC most common in Korea? Differences in Rx?

 Response: Thank you for your insightful comment. Small cell carcinoma accounted for only 13.5% of patients in the Korea Lung Cancer Registry (KALC-R) developed by the Korean Association for Lung Cancer and Korean Central Cancer Registry to produce unbiased and reliable epidemiological data (Kim et al., 2019). In addition, SCLC is an aggressive malignancy with a short doubling time, high fraction ratio, and early development of distant metastasis (Sabari et al., 2017). Since the treatment characteristics of SCLC and NSCLC are different, such as combined-modality treatment with etoposide and cisplatin with thoracic radiation therapy for SCLC, NCCN guidelines are being developed for each cancer separately. Thus, we included only non-small cell lung cancer in our study.

[References]

Young-Chul kim, Young-Joo Won. The Development of the Korean Lung Cancer Registry (KALC-R). Tuberc Respir Dis (Seoul). 2019; 82(2):91-93.

Sabari JK, Lok BH, Laird JH, Poirier JT, Rudin CM. Unravelling the biology of SCLC: implications for therapy. Nat Rev Clin Oncol. 2017;14:549-61.

Comment #4. If it was an interview based survey, participants needs not have to read Korean. Maybe replace with “communicate” in Korean.

 Response: We greatly appreciate the suggestions. We edited the word “read” to “communicate.”

 [Materials and Methods] (page 2, paragraph 6, line 89)

Cancer patients <18 years of age or not able to communicate in Korean were excluded.

Comment #5. Delete the sentence “considering the statistical significance level (P<0.1).

Response: Thank you for the comments. We apologize for the typo. P<0.05 is correct. We removed the sentence at your suggestion.

Comment #6.  Were these proportions reflective of national statistics in Korea? Are women and non-smokers diagnosed at such high proportions maybe related factors other than smoking such as indoor air pollution?

 Response: We appreciate the reviewer’s valuable comment. In a previous study in 2012, only 23.2% of the sample was women (Yun et al., 2013). Lung cancer revealed an increasing trend among cancers particularly in women. According to “Cancer Statistics in Korea: Incidence, Mortality, Survival, and Prevalence in 2019,” a total of 103,108 people were diagnosed with lung cancer in Korea in 2019, of which 60.2% (62,105 cases) were men and 39.8% (41,003 cases) were women (Kang et al., 2019). In our study, 55.7% were men and 44.3% were women, similar to the national statistics in Korea.

 In addition, according to a previous study using the Korean National Health Insurance Service database, which linked population data to Korea Central Cancer Registry data, 59.4% of 6,567,909 patients were non-smoking lung cancer patients (Moon et al, 2020), similar to the rate in our study (48.9%). Interestingly, never-smoker lung cancer patients are more likely to be female, have adenocarcinoma, and manifest with stage III or IV disease compared to lung cancer patients with a smoking history (Young-Chul et al., 2019). As never-smokers are not regarded as a high-risk group for lung cancer, the group is not screened by the national program. Thus, active research is needed to identify possible related factors other than smoking, such as indoor air pollution.

[References]

Yun YH, Shon EJ, Yang AJ, Kim SH, Kim YA, Chang YJ, et al. Needs regarding care and factors associated with unmet needs in disease-free survivors of surgically treated lung cancer. Ann Oncol. 2013;24:1552-9.

Kang MJ, Won YJ, Lee JJ, et al. Cancer statistics in Korea: Incidence, Mortality, Survival, and Prevalence in 2019. Cancer Rest Treat. 2022; 54(2):330-344.

Moon DH, Kwon SO, Kim SY, Kim WJ. Air poluution and Incidence of Lung Cancer by Histological Type in Korean Adults: A Korean National Health Insurance Service Health Examinee Cohort Study. Int J Enviorn Res Public Health. 2020;17(3):915

Young-Chul kim, Young-Joo Won. The Development of the Korean Lung Cancer Registry (KALC-R). Tuberc Respir Dis (Seoul). 2019; 82(2):91-93.

Comment #7. Were stage III patients free from metastases as listed in the exclusions? Please add this information in the foot notes.

 Response: We excluded patients with metastasis during study recruitment. We added this information in the Methods section and in the footnotes.

 [Materials and Methods] (page 2, paragraph 6, line 91)

Patients who did not complete treatment within at least 1 month before study participation, experienced recurrence/metastasis of cancer, or had secondary cancer were also excluded from the study.

Comment #8. A sensitivity analysis to check whether the findings differed by length of survival would be very helpful.

 Response: Thank you for your insightful comments. We performed a sensitivity analysis by length of survival dichotomized as <18 and ≥18 months (see the Supplementary Tables below). In both groups, high FCR was associated with moderate-to-strong unmet needs across all domains of CaSUN-K. Interestingly, poor social function was significant in those with a shorter survival duration (<18 months), while poor emotional function was significant in those with a longer survival duration (≥18 months), This suggests that survivors who have a physical condition or have received medical treatment that interfered with their social life have an increased risk of moderate-to-strong unmet needs during the shorter duration of survival. But, survivors gradually adapt over time, those with poor emotional function such as depressed are at risk of being moderate-to-strong unmet needs across all domains.

[Materials and Methods] (page 3, paragraph 6, line 136)

Stratified analyses were performed to determine the differences between lengths of survival.

[Results] (page 6, paragraph 1, line 175)

In a sensitivity analysis by length of survival dichotomized as <18 and ≥18 months (Supplementary Tables 1, 2), high FCR was associated with moderate-to-strong unmet needs across all domains. Interestingly, poor social function was significant in NSCLC patients with shorter survival duration (<18 months), while poor emotional function was significant in those with longer survival duration (≥18 months).

[Discussion] (page 12, paragraph 4, line 216)

Furthermore, high FCR remained significant in sensitivity analysis by length of survival. Survivors who have a physical condition or have received medical treatment that interfered with their social life had an increased risk of unmet needs across all domains in those with a shorter duration of survival. However, survivors gradually adapt over time, those with poor emotional function such as depressed are at risk of being moderate-to-strong unmet needs across all domains.

Supplementary Table 1. Predictor of moderate-to-strong level of unmet needs of CaSUN-K in those with shorter survival duration (< 18 months)

Information

Comprehensive Cancer Care

Existential Survivorship

Quality of Life

Relationship

Financial issues

OR

(95% CI)

aOR*

(95% CI)

OR

(95% CI)

aOR*

(95% CI)

OR

(95% CI)

aOR*

(95% CI)

OR

(95% CI)

aOR*

(95% CI)

OR

(95% CI)

aOR*

(95% CI)

OR

(95% CI)

aOR*

(95% CI)

Socio-demographic characteristics

Age

≥ 65 years

1.00

1.00

1.00

1.00

1.00

1.00

< 65 years

1.22

(0.84, 1.76)

0.93

(0.65, 1.35)

1.09

(0.74, 1.60)

1.03

(0.71, 1.49)

0.94

(0.64, 1.38)

1.49

(0.98, 2.28)

Sex

Female

1.00

1.00

1.00

1.00

1.00

1.00

Male

0.80

(0.55, 1.16)

1.09

(0.75, 1.57)

0.72

(0.49, 1.05)

0.96

(0.66, 1.39)

0.97

(0.66, 0.72)

0.90

(0.60, 1.36)

Marital status

Non-married

1.00

1.00

1.00

1.00

1.00

1.00

Married

0.69

(0.40, 1.20)

1.29

(0.76, 2.20)

0.61

(0.41, 0.91)

0.80

(0.47, 1.36)

0.88

(0.51, 1.52)

0.76

(0.43, 1.35)

Education level

≥ University

1.00

1.00

1.00

1.00

1.00

1.00

High school

1.15

(0.74, 1.78)

1.09

(0.70, 1.70)

0.99

(0.63, 1.57)

0.99

(0.64, 1.53)

1.13

(0.71, 1.78)

1.14

(0.70, 1.85)

≤ Middle school

1.18

(0.75, 1.86)

1.01

(0.64, 1.60)

0.90

(0.56, 1.46)

0.89

(0.57, 1.41)

1.00

(0.62, 1.62)

0.74

(0.44, 1.27)

Employment status

  Employed

1.00

1.00

1.00

1.00

1.00

1.00

  Unemployed

1.21

(0.84, 1.76)

1.10

(0.75, 1.59)

1.56

(1.06, 2.30)

1.12

(0.77, 1.62)

1.43

(0.97, 2.13)

0.75

(0.50, 1.14)

0.55

(0.31, 0.96)

Household monthly income

  < USD$3,000

1.00

1.00

1.00

1.00

1.00

1.00

≥ UDS$3,000

0.87

(0.59, 1.29)

0.98

(0.66, 1.45)

0.91

(0.60, 1.37)

1.66

(1.03, 2.68)

0.98

(0.66, 1.45)

0.74

(0.49, 1.12)

1.09

(0.69, 1.71)

Disease-related characteristics

  Comorbidity

    No

1.00

1.00

1.00

1.00

1.00

1.00

  Yes

1.33

(0.87, 2.03)

1.57

(1.02, 2.40)

1.28

(0.82, 1.98)

1.72

(1.11, 2.67)

1.72

(1.02, 2.89)

1.20

(0.76, 1.88)

0.74

(0.47, 1.18)

0.55

(0.32, 0.96)

Pathologic stage

0-I

1.00

1.00

1.00

1.00

1.00

1.00

II

1.06

(0.63, 1.80)

1.09

(0.64, 1.85)

1.21

(0.69, 2.11)

1.46

(0.87, 2.48)

1.39

(0.81, 2.38)

1.45

(0.82, 2.58)

III

2.44

(1.30, 4.59)

2.22

(1.18, 4.17)

2.12

(1.08, 4.17)

2.49

(1.38, 4.49)

1.48

(0.83, 2.64)

1.82

(1.00, 3.33)

Psychological characteristics

Fear of Cancer Recurrence

Low FCR (< 13)

1.00

1.00

1.00

1.00

1.00

1.00

High FCR (≥ 13)

5.13

(3.44, 7.67)

4.93

(3.16, 7.71)

4.61

(3.10, 6.86)

4.26

(2.74, 6.63)

7.81

(5.06, 12.04)

6.73

(4.19, 10.83)

5.34

(3.52, 8.09)

4.68

(2.87, 7.61)

6.43

(3.99, 10.37)

5.81

(3.40, 9.91)

4.28

(2.59, 7.06)

3.74

(2.12, 6.60)

EORTC-QoL-C30

Poor Physical Function

2.67

(1.64, 4.34)

1.85

(1.04, 3.29)

2.11

(1.31, 3.41)

3.43

(1.93, 6.09)

2.20

(1.40, 3.46)

1.77

(1.12, 2.78)

1.63

(1.01, 2.63)

Poor Role Function

2.44

(1.67, 3.57)

2.32

(1.59, 3.39)

2.32

(1.56, 3.45)

2.08

(1.43, 3.02)

2.18

(1.48, 3.23)

2.32

(1.52, 3.55)

Poor Emotional Function

3.36

(2.22, 5.09)

3.21

(2.12, 4.88)

3.89

(2.44, 6.20)

2.92

(1.97, 4.33)

1.93

(1.14, 3.25)

3.71

(2.47, 5.57)

3.34

(2.18, 5.14)

Poor Cognitive Function

2.99

(1.89, 4.75)

1.81

(1.05, 3.13)

3.16

(1.97, 5.07)

1.97

(1.13, 3.45)

2.94

(1.76, 4.90)

2.59

(1.68, 3.99)

3.01

(1.95, 4.63)

1.70

(1.02, 2.82)

2.92

(1.86, 4.57)

2.28

(1.34, 3.88)

    Poor Social Function

2.65

(1.80, 3.91)

1.62

(1.01, 2.60)

3.15

(2.11, 4.69)

2.46

(1.54, 3.93)

4.48

(2.86, 7.01)

3.60

(2.14, 6.04)

2.63

(1.79, 3.84)

2.10

(1.29, 3.42)

3.06

(2.06, 4.55)

2.17

(1.36, 3.45)

2.79

(1.82, 4.27)

1.96

(1.19, 3.21)

Abbreviation: CaSUN-K= Korean version of the Cancer Survivors’ Unmet Needs survey, EORTC-QoL-C30=European organization for research and treatment of cancer quality of life core questionnaire, OR=odds ratio, aOR=adjusted odds ratio, CI=confidence interval. *Data are expressed based on multivariable logistic models by stepwise backward selection considering significance at p<0.05

Supplementary Table 2. Predictors of moderate-to-strong level of unmet need of CaSUN-K by domain in those with longer survival duration (≥ 18 months)

Information

Comprehensive Cancer Care

Existential Survivorship

Quality of Life

Relationship

Financial issues

OR

(95% CI)

aOR*

(95% CI)

OR

(95% CI)

aOR*

(95% CI)

OR

(95% CI)

aOR*

(95% CI)

OR

(95% CI)

aOR*

(95% CI)

OR

(95% CI)

aOR*

(95% CI)

OR

(95% CI)

aOR*

(95% CI)

Socio-demographic characteristics

Age

≥ 65 years

1.00

1.00

1.00

1.00

1.00

1.00

< 65 years

0.97

(0.68, 1.39)

1.22

(0.85, 1.75)

1.29

(0.90, 1.84)

1.35

(0.92, 1.98)

1.47

(0.98, 2.20)

1.44

(0.91, 2.29)

1.78

(1.05, 3.01)

Sex

Female

1.00

1.00

1.00

1.00

1.00

1.00

Male

0.99

(0.69, 1.42)

1.20

(0.83, 1.72)

0.86

(0.60, 1.24)

0.74

(0.51, 1.09)

1.09

(0.72, 1.64)

1.00

(0.63, 1.59)

Marital status

Non-married

1.00

1.00

1.00

1.00

1.00

1.00

Married

0.94

(0.56, 1.59)

1.22

(0.72, 2.06)

0.73

(0.43, 1.24)

0.70

(0.41, 1.19)

1.29

(0.70, 2.39)

0.61

(0.33, 1.12)

Education level

≥ University

1.00

1.00

1.00

1.00

1.00

1.00

High school

0.86

(0.56, 1.30)

1.08

(0.71, 1.64)

1.12

(0.74, 1.70)

1.00

(0.64, 1.56)

0.74

(0.46, 1.18)

0.85

(0.50, 1.45)

≤ Middle school

0.98

(0.62, 1.53)

1.04

(0.66, 1.63)

1.11

(0.71, 1.74)

1.19

(0.74, 1.91)

0.70

(0.42, 1.15)

0.97

(0.55, 1.71)

Employment status

  Employed

1.00

1.00

1.00

1.00

1.00

1.00

  Unemployed

0.95

(0.66, 1.36)

0.74

(0.52, 1.06)

0.61

(0.41, 0.93)

1.10

(0.77, 1.57)

1.06

(0.73, 1.56)

0.92

(0.62, 1.38)

1.31

(0.82, 2.09)

Household monthly income

  < USD$3,000

1.00

1.00

1.00

1.00

1.00

1.00

≥ USD$3,000

0.95

(0.65, 1.38)

1.08

(0.74, 1.58)

0.71

(0.48, 1.04)

0.88

(0.59, 1.31)

1.10

(0.72, 1.69)

0.51

(0.32, 0.82)

0.48

(0.28, 0.80)

Disease-related characteristics

  Comorbidity

    No

1.00

1.00

1.00

1.00

1.00

1.00

  Yes

0.94

(0.63, 1.41)

0.94

(0.63, 1.41)

1.01

(0.67, 1.51)

0.92

(0.60, 1.42)

0.92

(0.58, 1.44)

0.86

(0.52, 1.42)

Pathologic stage

0-I

1.00

1.00

1.00

1.00

1.00

1.00

II

1.44

(0.87, 2.38)

1.86

(1.05, 3.29)

1.08

(0.66, 1.79)

1.26

(0.76, 2.09)

1.29

(0.76, 2.17)

1.56

(0.91, 2.65)

1.94

(1.07, 3.52)

1.19

(0.64, 2.23)

III

0.69

(0.36, 1.32)

0.78

(0.42, 1.47)

1.27

(0.68, 2.37)

1.33

(0.70, 2.52)

0.80

(0.38, 0.45)

1.49

(0.72, 3.09)

Psychological characteristics

Fear of Cancer Recurrence

Low FCR (< 13)

1.00

1.00

1.00

1.00

1.00

1.00

High FCR (≥ 13)

4.17

(2.84, 6.13)

4.03

(2.62, 6.20)

3.97

(2.71, 5.81)

4.00

(2.62, 6.09)

6.51

(4.38, 9.66)

6.17

(3.98, 9.56)

4.58

(2.98, 7.02)

3.90

(2.79, 5.46)

4.52

(2.85, 7.18)

4.19

(2.51, 6.99)

3.42

(2.03, 5.73)

2.48

(1.41, 4.35)

EORTC-QoL-C30

Poor Physical Function

1.73

(1.03, 2.89)

1.55

(0.93, 2.59)

2.62

(1.49, 4.59)

1.83

(1.09, 3.08)

1.64

(0.96, 2.80)

1.65

(0.91, 2.98)

Poor Role Function

1.87

(1.28, 2.74)

1.65

(1.13, 2.41)

1.88

(1.28, 2.77)

2.09

(1.41, 3.09)

2.27

(1.51, 3.42)

2.40

(1.51, 3.81)

Poor Emotional Function

3.40

(2.22, 5.19)

2.05

(1.24, 3.38)

2.82

(2.86, 4.29)

2.01

(1.26, 3.20)

3.84

(2.44, 6.05)

2.09

(1.21, 3.62)

4.08

(2.67, 6.23)

1.86

(1.30, 2.65)

3.41

(2.21, 5.24)

2.48

(1.55, 3.98)

4.00

(2.49, 6.43)

2.78

(1.66, 4.65)

Poor Cognitive Function

2.43

(1.61, 3.69)

1.95

(1.29, 2.94)

2.56

(1.66, 3.94)

1.90

(1.25, 2.88)

1.82

(1.18, 2.81)

1.94

(1.19, 3.14)

    Poor Social Function

2.62

(1.75, 3.93)

2.40

(1.60, 3.58)

3.61

(2.34, 5.57)

1.96

(1.16, 3.34)

3.83

(2.53, 5.78)

2.00

(1.42, 2.80)

2.78

(1.82, 4.24)

2.12

(1.32, 3.40)

Abbreviation: CaSUN-K= Korean version of the Cancer Survivors’ Unmet Needs survey, EORTC-QoL-C30=European organization for research and treatment of cancer quality of life core questionnaire, OR=odds ratio, aOR=adjusted odds ratio, CI=confidence interval. *Data are expressed based on multivariable logistic models by stepwise backward selection considering significance at p<0.05

Comment #9. This is in contrast to Yun et al study cited in the introduction. Exploring the results more would be helpful. I would have expected that women might have more unmet needs than men as they would be the primary care taker for their spouse/partner. This may not be true for women patients. Unless the findings in Korea might be different.

 Response: Thank you for the critical comment. Yun et al. reported female as a predictor of unmet needs. However, there was no significant difference in their tables (see Tables 2 and 3 below). In that study, a validated instrument for assessing unmet needs was not used, and psychological factors known to be associated with unmet needs such as QoL were not evaluated. Interestingly, our result showed that male sex was not significant before adjustment in univariable regression analysis. However, male showed an increased risk of unmet need in the comprehensive cancer care domain after adjustment for confounding factors including psychological factors such as FCR and functioning on the EORTC QLQ-C30. Thus, different results could be obtained depending on the adjusted confounding variables. Further research should clarify this complex relationship.

Figure 1. Factors associated with high unmet need (Yun et al, Ann Oncol, 2013)

Figure 2. Predictors of high unmet need (Yun et al, Ann Oncol, 2013)

[References]

Yun YH, Shon EJ, Yang AJ, Kim SH, Kim YA, Chang YJ, et al. Needs regarding care and factors associated with unmet needs in disease-free survivors of surgically treated lung cancer. Ann Oncol. 2013;24:1552-9.

Reviewer 2 Report

This is a well-written manuscript detailing a significant investigation into the level and correlates of unmet supportive care needs among NSCLC survivors in South Korea. The main issue with this paper is that univariable logistic regression was not conducted. Both univariable and multivariable logistic regression analysis need to be conducted, so as readers can see how odds ratios and corresponding 95% confidence intervals change after adjustment. A further issue is that there is no indication of the extent of missing data and how missing data were handled in the analysis. If there were no missing data, then a response rate of 100% should be given. Additionally, minor updates to the manuscript are required for the sake of clarity and correctness, as per the attached version of the manuscript.

The quality of the English language in this manuscript is very high.

Author Response

Response to comments from the Reviewer 2

Comment #1. I’d suggest extending the title to include the city and country this study was conducted in: Seoul, South Korea.

Response: Thank you for the comment. We extended the title to, “Unmet supportive care needs after non-small cell lung cancer resection at a tertiary hospital in Seoul, South Korea.”

[Title] (page 1, line 2)

Unmet supportive care needs after non-small cell lung cancer resection at a tertiary hospital in Seoul, South Korea

Comment #2. I’d suggest extending this to ‘in Seoul, South Korea’/

Response: Thank you for the comment. We updated the sentence ‘in Korea’ to ‘in Seoul, South Korea’ as suggested.

[Abstract] (page 1, line 23)

The aim of this study is to identify the prevalence and predictors of unmet needs of non-small cell lung cancer (NSCLC) patients undergoing surgical resection in Seoul, South Korea.

Comment #3. Inset the apostrophe after the s.

Response: Thank you for the comment. We changed “survivor’s” to “survivors’.”

[Abstract] (page 1, line 24)

A total of 949 patients who completed survey questionnaires that included the Cancer Survivors’ Unmet Needs Korean version (CaSUN-K), fear of cancer recurrence (FCR) inventory-short form, and European Organization for Research and Treatment of Cancer Quality of Life Core Questionnaire (EORTC QLQ-C30) was recruited from January to October 2020.

Comment #4. The word ‘correlates’ would be more appropriate to use, given the multivariable logistic regression model is an explanatory rather than predictive model.

Response: Thank you very much for the insightful comments. We changed “predictors” to “correlates” as suggested.

[Abstract] (page 1, line 28)

Multivariable logistic regression was used to determine potential correlation of significant unmet need, defined as any moderate or strong need, for each domain of CaSUN-K.

Comment #5. Insert the sample size before describing the sample.

Response: We appreciate the reviewer’s thoughtful comments. We included the sample size of this study.

[Abstract] (page 1, line 29)

Of the 949 participants, the mean age was 63.4 ± 8.8 years and 529 (55.7%) were male.

Comment #6. State the exact percentage in words rather than being imprecise.

Response: We changed ‘more than 70%’ to ’91.8%’ Thank you for the comment.

 [Abstract] (page 1, line 30)

91.8% of participants reported one or more unmet need.

Comment #7. It would be ideal to highlight other significant associations too

Response: We appreciate the reviewer’s comments. We agree that it is ideal to suggest other significant associations. However, due to the word count limit for the Abstract (200 words), we presented other significant associations in the Results and Discussion instead. We included in the Abstract only the factors associated with moderate-to-strong unmet needs across all domains of CaSUN-K. Thus, we emphasize in the Conclusion that relief of FCR and impaired emotional functioning is important.

Comment #8. Replace lung cancer with NSCLC

Response: We edited ‘lung cancer survivors’ to ‘non-small cell lung cancer survivors.’

[Abstract] (page 1, line 34)

Non-small cell lung cancer survivors with a recent diagnosis had more frequent disease-related unmet needs. Interventions to reduce unmet needs of NSCLC patients should focused on relieving FCR and improving emotional functioning.

Comment #9. Change the ‘emphasis should be on’ to ‘emphasis should be placed on’.

Response: We edited the sentence. Thank you very much for the comment.

[Abstract] (page 1, line 36)

Furthermore, the emphasis should be placed on decreasing disease-related needs, particularly for early survivors of lung cancer during the re-entry phase.

Comment #10. Insert the apostrophe after s

Response: Thank you for your comment. We inserted the apostrophe as appropriate.

[Introduction] (page 2, paragraph 3, line 62)

Among the studies, the unmet needs based on the Cancer Survivors’ Unmet Needs (CaSUN) scale were investigated in only 1 study, and the moderate-to-strong unmet needs of various cancer survivors, including lung cancer, in Asian countries were reported.

Comment #11. Change to ‘correlates’, as per the above comment.

Response: We edited ‘predictors’ to ‘correlates.’ Thank you very much for the comment.

[Introduction] (page 2, paragraph 3, line 64)

However, the authors only reported results collected from the unmet needs survey and did not perform any further analysis such as exploring correlates of unmet needs in each domain.

[Introduction] (page 2, paragraph 4, line 69)

In Korea, Yun et al. previously attempted to determine significant correlates of unmet needs in Korean lung cancer survivors including factors of socioeconomic burden (female, employment, and fewer family members) and medical burden (chemotherapy and long disease duration after cancer diagnosis).

[Introduction] (page 2, paragraph 5, line 75)

Therefore, in the present study, the prevalence of unmet needs of lung cancer survivors was evaluated, and the correlates, including demographic, disease-related, and psychological factors, associated with unmet needs in different domains were explored.

Comment #12. Extend this to lung cancer survivors’

Response: We edited the word ‘survivors’ to ‘lung cancer survivors’. Thank you very much for the comment.

[Introduction] (page 2, paragraph 5, line 79)

The results can provide detailed information regarding the main correlates of unmet needs and show how NSCLC survivors’ unmet needs can be best addressed.

Comment #13. State the study design. This appears to be a cross-sectional study. State the sample size in the results rather than in the method. Specify the type of recruitment. Was this a convenience sample? Provide justifications for the inclusion and exclusion criteria.

Response: Thank you for the insightful comment. Participants were recruited using convenience sampling from thoracic surgery and long-term survivorship clinics in the Comprehensive Cancer Center at Samsung Medical Center. Trained researchers approached the patients at the outpatient clinic and provided detailed information about the study, after which 1,014 (83.1%) provided informed consent to participate. Patients who had previously experienced second cancer or cancer recurrence were excluded to avoid bias. We added this in the limitation section.

[Materials and Methods] (page 2, paragraph 6, line 83)

This cross-sectional study recruited 1,220 patients using convenience sampling from the outpatient clinics of thoracic surgery and long-term survivorship at the Samsung Comprehensive Cancer Center in South Korea from January to October 2020. Among them, 1,014 (83.1%) provided informed consent to participate the study

[Limitation] (page 13, paragraph 6, line 284)

The present study had several limitations. First, the study cohort might not represent the general NSCLC survivor population because it included only patients from one tertiary hospital recruited using convenience sampling. In addition, patients who had previously experienced second cancer or cancer recurrence/metastasis were excluded. Therefore, a selection bias may limit generalization of these findings to other NSCLC survivors

Comment #14. The word cancer is used in the abstract, rather than cancer’s. Please check.

Response: We edited the word ‘cancer’s’ to ‘cancer.’ Thank you very much for the comment.

[Materials and Methods] (page 3, paragraph 1, line 101)

A trained research assistant conducted a face-to-face interview to complete a questionnaire that included the CaSUN Korean version (CaSUN-K), the Korean version of the FCR (K-FCRI-SF) Inventory-Short Form, and the Korean version of European Organization for Research and Treatment of Cancer Quality of Life Core Questionnaire (EORTC QLQ-C30).

Comment #15. The below numbers of items don’t sum to 35. Please check.

Response: We appreciate your comment. The original CaSUN consists of 35 unmet needs items, with 28 items covering five domains. The remaining 7 items originally were not grouped into any domain. Therefore, we distributed these 7 items into appropriate domains.

Comment #16 It does not make sense to say ‘needs were needed’. I’d suggest updating this sentence to something like: ‘For each item, participants answer whether or not they had a particular need and, if they did, whether or not that need was met or unmet’.

Response: Thank you for your comment. We updated the sentence as you suggested.

[Materials and Methods] (page 3, paragraph 2, line 108)

For each item, participants answer whether or not they had a particular need and, if they did, whether or not that need was met or unmet.

Comment #17 The word ‘rate’ should be ‘rated’. Insert the word them here.

Response: Thank you for the comment. We updated the word as suggested.

[Materials and Methods] (page 3, paragraph 2, line 110)

If an unmet need was reported, the participant rated the intensity of the need as weak, moderate, or strong. In the present study, moderate-to-strong ratings of unmet needs were discussed further based on several previous studies.

[Materials and Methods] (page 3, paragraph 3, line 116)

In addition, 7 items that were originally not grouped into any dimension were distributed; 4 of them were classified into the financial dimension in CaSUN-K.

Comment #18 Provide the justification given in the past study for this clinical cut-off (13, FCR). Provide the justifications given for this cut-off in previous studies (66.7, EORTC-QLQ-C30).

Response: We appreciate your valuable comment. Simard et al. randomly selected survivors of localized breast, prostate, lung, or colorectal cancer who had been treated within the previous 4 years and suggested a cut-off of 13 on the FCRI-SF, with optimal sensitivity (88%) and specificity (75%) rates for the screening of clinical levels of FCR.

In addition, when using the European Organization for Research Treatment of Cancer Quality of Life Questionnaire C30 to identify people with psychological distress, a cut-off of <66.7 is often used based on the population distribution of scores. Several previous studies reported use of this cut-off in breast, colon, and melanoma cancer patients. Nevertheless, there is no consensus on the standard cut-off point for subjects with cancer. Some authors have found a lower mean score in cancer patients, while others recommend using a cut-off point <0.75 or even <90. Therefore, we listed this point as a limitation of our study in the Discussion.

[References]

Simard S, Savard J. Screening and comorbidity of clinical levels of fear of cancer recurrence. J Cancer Surviv. 2015;9(3):481-491.

Calderon C, Carmona-Bayonas A, Jara C, et al. Emotional functioning to screen for pscyhological distress in breast and colorectal cancer patients prior to adjuvant treatment initiation. Eur J Cancer Care (Engl). 2019;28(3):e13005.

Aamdal E, Skouvlund E, Jacobsen KD, et al. Health-related quality of life in patients with advanced melanoma treated with ipilimumab: prognostic implications and changes during treatment. ESMO Open. 2022;7(5):100588.

Giuliani ME, Miline RA, Puts M, Sampson LR, Kwan JY, Le LW, et al. The prevalence an dnature of supportive care needs in lung cacner patients. Curr Oncol. 2016;23:258-65.

[Limitation] (page 13, paragraph 6, line 290)

Third, there is no consensus on the standard cut-off point of the EORTC-QLQ-C30 with cancer patients. A cut-off point <66.7 was used in the present study based on the population distribution of scores.

Comment #19 Replace ‘identify’ with ‘describe’. At the end of this sentence, insert as well as all other variables.

Response: Thank you for the comment. We updated the word as suggested.

[Materials and Methods] (page 3, paragraph 6, line 129)

Descriptive statistics were used to describe the domains of unmet needs as well as all other variables.

Comment #20 Extend multivariable to univariable and multivariable. Univariable logistic regression analysis needs to be conducted too, so as readers can see how odds ratios and corresponding 95% confidence intervals change after adjustment.

Response: We appreciate your thoughtful comment. We conducted univariable analysis to help our readers understand the result.

[Materials and Methods] (page 3, paragraph 6, line 130)

Univariable and multivariable logistic regression analyses were used to analyze the potential correlates of each domain of the unmet need

Table 3. Predictors of moderate-to-strong level of unmet needs of CaSUN-K by domain by uni- and multivariable logistic regression

Information

Comprehensive Cancer Care

Existential Survivorship

Quality of Life

Relationship

Financial issues

OR

(95% CI)

aOR*

(95% CI)

OR

(95% CI)

aOR*

(95% CI)

OR

(95% CI)

aOR*

(95% CI)

OR

(95% CI)

aOR*

(95% CI)

OR

(95% CI)

aOR*

(95% CI)

OR

(95% CI)

aOR*

(95% CI)

Socio-demographic characteristics

Age

≥ 65 years

1.00

1.00

1.00

1.00

1.00

1.00

< 65 years

1.22

(0.84, 1.76)

0.93

(0.65, 1.35)

1.09

(0.74, 1.60)

1.03

(0.71, 1.49)

0.94

(0.64, 1.38)

1.49

(0.98, 2.28)

Sex

Female

1.00

1.00

1.00

1.00

1.00

1.00

Male

0.80

(0.55, 1.16)

1.09

(0.75, 1.57)

0.72

(0.49, 1.05)

0.96

(0.66, 1.39)

0.97

(0.66, 0.72)

0.90

(0.60, 1.36)

Marital status

Non-married

1.00

1.00

1.00

1.00

1.00

1.00

Married

0.69

(0.40, 1.20)

1.29

(0.76, 2.20)

0.61

(0.41, 0.91)

0.80

(0.47, 1.36)

0.88

(0.51, 1.52)

0.76

(0.43, 1.35)

Education level

≥ University

1.00

1.00

1.00

1.00

1.00

1.00

High school

1.15

(0.74, 1.78)

1.09

(0.70, 1.70)

0.99

(0.63, 1.57)

0.99

(0.64, 1.53)

1.13

(0.71, 1.78)

1.14

(0.70, 1.85)

≤ Middle school

1.18

(0.75, 1.86)

1.01

(0.64, 1.60)

0.90

(0.56, 1.46)

0.89

(0.57, 1.41)

1.00

(0.62, 1.62)

0.74

(0.44, 1.27)

Employment status

  Employed

1.00

1.00

1.00

1.00

1.00

1.00

  Unemployed

1.21

(0.84, 1.76)

1.10

(0.75, 1.59)

1.56

(1.06, 2.30)

1.12

(0.77, 1.62)

1.43

(0.97, 2.13)

0.75

(0.50, 1.14)

0.55

(0.31, 0.96)

Household monthly income

  < USD$3,000

1.00

1.00

1.00

1.00

1.00

1.00

≥ USD$3,000

0.87

(0.59, 1.29)

0.98

(0.66, 1.45)

0.91

(0.60, 1.37)

1.66

(1.03, 2.68)

0.98

(0.66, 1.45)

0.74

(0.49, 1.12)

1.09

(0.69, 1.71)

Disease-related characteristics

  Comorbidity

    No

1.00

1.00

1.00

1.00

1.00

1.00

  Yes

1.33

(0.87, 2.03)

1.57

(1.02, 2.40)

1.28

(0.82, 1.98)

1.72

(1.11, 2.67)

1.72

(1.02, 2.89)

1.20

(0.76, 1.88)

0.74

(0.47, 1.18)

0.55

(0.32, 0.96)

Pathologic stage

0-I

1.00

1.00

1.00

1.00

1.00

1.00

II

1.06

(0.63, 1.80)

1.09

(0.64, 1.85)

1.21

(0.69, 2.11)

1.46

(0.87, 2.48)

1.39

(0.81, 2.38)

1.45

(0.82, 2.58)

III

2.44

(1.30, 4.59)

2.22

(1.18, 4.17)

2.12

(1.08, 4.17)

2.49

(1.38, 4.49)

1.48

(0.83, 2.64)

1.82

(1.00, 3.33)

Psychological characteristics

Fear of Cancer Recurrence

Low FCR (< 13)

1.00

1.00

1.00

1.00

1.00

1.00

High FCR (≥ 13)

5.13

(3.44, 7.67)

4.93

(3.16, 7.71)

4.61

(3.10, 6.86)

4.26

(2.74, 6.63)

7.81

(5.06, 12.04)

6.73

(4.19, 10.83)

5.34

(3.52, 8.09)

4.68

(2.87, 7.61)

6.43

(3.99, 10.37)

5.81

(3.40, 9.91)

4.28

(2.59, 7.06)

3.74

(2.12, 6.60)

EORTC-QoL-C30

Poor Physical Function

2.67

(1.64, 4.34)

1.85

(1.04, 3.29)

2.11

(1.31, 3.41)

3.43

(1.93, 6.09)

2.20

(1.40, 3.46)

1.77

(1.12, 2.78)

1.63

(1.01, 2.63)

Poor Role Function

2.44

(1.67, 3.57)

2.32

(1.59, 3.39)

2.32

(1.56, 3.45)

2.08

(1.43, 3.02)

2.18

(1.48, 3.23)

2.32

(1.52, 3.55)

Poor Emotional Function

3.36

(2.22, 5.09)

3.21

(2.12, 4.88)

3.89

(2.44, 6.20)

2.92

(1.97, 4.33)

1.93

(1.14, 3.25)

3.71

(2.47, 5.57)

3.34

(2.18, 5.14)

Poor Cognitive Function

2.99

(1.89, 4.75)

1.81

(1.05, 3.13)

3.16

(1.97, 5.07)

1.97

(1.13, 3.45)

2.94

(1.76, 4.90)

2.59

(1.68, 3.99)

3.01

(1.95, 4.63)

1.70

(1.02, 2.82)

2.92

(1.86, 4.57)

2.28

(1.34, 3.88)

    Poor Social Function

2.65

(1.80, 3.91)

1.62

(1.01, 2.60)

3.15

(2.11, 4.69)

2.46

(1.54, 3.93)

4.48

(2.86, 7.01)

3.60

(2.14, 6.04)

2.63

(1.79, 3.84)

2.10

(1.29, 3.42)

3.06

(2.06, 4.55)

2.17

(1.36, 3.45)

2.79

(1.82, 4.27)

1.96

(1.19, 3.21)

Abbreviation: CaSUN-K= Korean version of the Cancer Survivors’ Unmet Needs, EORTC-QoL-C30=European organization for research and treatment of cancer quality of life core questionnaire, OR=odds ratio, aOR=adjusted odds ratio, CI=confidence interval. *Data are expressed based on multivariable logistic models by stepwise backward selection considering significance at p<0.05

Comment #21 Provide relevant references for these known confounding factors.

Response: Previously, Yun et al. demonstrated several significant correlates of unmet needs in Korean lung cancer survivors, including socioeconomic burden (female, employment, and fewer family members) and medical burden (chemotherapy and long disease duration after cancer diagnosis) (Yun et al., 2013). Furthermore, a recent systematic review of unmet supportive care needs in lung cancer patients investigated confounding factors (Cochrane et al, 2022; Molassiotis et al, 2017).

[References]

Yun YH, Shon EJ, Yang AJ, Kim SH, Kim YA, Chang YJ, et al. Needs regarding care and factors associated with unmet needs in disease-free survivors of surgically treated lung cancer. Ann Oncol. 2013;24:1552-9.

Cochrane A, Woods S, Dunne S, Gallagher P. Unmet supportive care needs associated with quality of life for people with lung cancer: A systematic review of the evidence 2007-2020. Eur J Cancer Care (Engl). 2022;31:e13525.

Molassiotis A, Yates P, Li Q, So WKW, Pongthavornkamol K, Pittayapan P, et al. Mapping unmet supportive care needs, quality-of-life perceptions and current symptoms in cancer survivors across the Asia-Pacific region: results from the International STEP Study. Ann Oncol. 2017;28:2552-8.

Comment #22 Provide a reference and/or justification for this model building approach.

Response: Thank you for the comments. We apologize for the typo. P<0.05 is correct. We removed the sentence at Reviewer 1’s suggestion. Please refer to the Reviewer 1’s Comment #5.

Comment #23 Replace this with stata/MP version 14.0 (Stata Corp, College station, Texas, USA).

Response: Thank you for the comment. We updated the word as suggested.

[Materials and Methods] (page 3, paragraph 6, line 137)

All statistical analyses were conducted using STATA/MP 14.0 (Stata Corp, College Station, TX, USA).

Comment #24 A subsection on ethical consideration is required, including the relevant ethics committee and ID number.

Response: Thank you for the comment. We reported the ethical considerations including IRB number after the funding section.

[Institutional Review Board Statement] (page 14, paragraph 6, line 323)

The Institutional Review Board (IRB) of Samsung Medical Center (SMC 2018-09-037) reviewed and approved this study. The study complied with the ethical rules for human experimentation described in the Declaration of Helsinki.

Comment #25 What was the response rate for all the questionnaires? How much data were missing and how were missing data handled in the analysis? If there were no missing data, then state that the response rate was 100%.

Response: Thank you for the insightful comment. Among the 1,220 patients invited to participate in the study, we received informed consent from 1,014 (83.1%). After applying the exclusion criteria, 949 patients were included in the final analysis. Trained researchers approached the patients at the outpatient clinic and conducted a face-to-face interview to complete a questionnaire. Thus, there were few missing data points for questionnaires except educational level (n=1) and household income (n=78), as we reported in Table 1. The missing data were handled as valid in the analysis.

[Materials and Methods] (page 2, paragraph 6, line 83)

This cross-sectional study recruited 1,220 patients using convenience sampling from the outpatient clinics of thoracic surgery and long-term survivorship at the Samsung Comprehensive Cancer Center in South Korea from January to October 2020. Among them, 1,014 (83.1%) provided informed consent to participate the study

Comment #26 Insert the sample size before describing the sample.

Response: Thank you for the comment. We added the sample size.

[Results] (page 3, paragraph 7, line 141)

Among the total 949 participants, the mean age was 63.4 ± 8.8 years old and 529 (55.7%) were male. Among them, 48.9% were never smokers and 41.9% were employed.

Comment #27 Extend ‘in’ to ‘in terms of’. 40.5% should be 51.0%.

Response: Thank you for the comment. There was an error in the number of subjects. We updated the number from 484 to 384 (40.5%).

Comment #28 Given rounding and calculation errors are evident in this table, as per the below comment, I would suggest checking all rounding and calculations in other tables and figures as well as the text. Add the following column heading: Characteristics. State what is in this column in the column heading (i.e. frequency (percentage), unless otherwise indicated).

Response: Thank you for the comment. We modified Table 1 as below.

Table 1. Baseline characteristics of the total study population

Characteristics

Total study population (n=949)

Number (%)

Socio-demographic characteristics

Mean age, years*

63.4 ± 8.8

< 65

510 (53.7)

≥ 65

439 (46.3)

Sex

Male

529 (55.7)

Female

420 (44.3)

Smoking status

Never smoker

464 (48.9)

Ex- or current smoker

485 (51.1)

Education level

≥ University

353 (37.2)

High school

328 (34.6)

≤ Middle school

267 (28.1)

Unknown

1 (0.1)

Marital status

Married

820 (86.4)

Working status

    Unemployed

551 (58.1)

Employed

398 (41.9)

Household monthly income

< USD$3,000

335 (35.3)

≥ USD$3,000

536 (56.5)

Unknown

78 (8.2)

Disease-related characteristics

Comorbidity

No

239 (25.2)

Yes

710 (74.8)

Pathologic stage

0

12 (1.3)

I

697 (73.4)

â…¡

140 (14.8)

â…¢

100 (10.5)

Time since the end of active treatment

< 18 months

461 (48.6)

≥ 18 months

488 (51.4)

Psychological characteristics

Fear of cancer recurrence (FCR)

Low FCR (< 13)

419 (44.2)

High FCR (≥ 13)

530 (55.8)

Function domain of EORTC-QoL-C30

Poor Physical Function

168 (17.7)

Poor Role Function

384 (51.0)

Poor Emotional Function

293 (30.9)

Poor Cognitive Function

244 (25.7)

Poor Social Function

332 (35.0)

Abbreviation: EORTC-QoL-C30=European organization for research and treatment of cancer quality of life core questionnaire. *Data are expressed as mean ± standard deviation.

Comment #30 Add a y-axis with the below line beginning proportion… as the y-axis label. Shift this up to the y-axis, as per the above comment. A legend is not required here.

Response: We appreciate your valuable comment. Figure 1 was modified as below.

Comment #31. Throughout this paragraph, change were predicted based on to were associated with its important to use the word associated because the direction of the association is uncertain in a cross-sectional study. One cannot claim that on variable predicts the other. There’s no need to restate all the aORs and 95%CIs, given these are in the table.

Response: We appreciate your comment. We changed ‘predicted’ to ‘associated with’ throughout the paragraph and deleted the aORs and 95%CIs.

[Results] (page 6, paragraph 1, line 166)

In Table 3, information needs were associated with high FCR , poor emotional function, and poor cognitive function. Comprehensive cancer care needs were associated with male gender, shorter time since the end of active treatment, high FCR, poor emotional function, and poor social function. Existential survivorship needs were associated with shorter time since the end of active treatment, high FCR , poor physical function, poor emotional function, and poor social function. QoL needs were associated with shorter time since the end of active treatment, high FCR, poor emotional function, and poor social function. Relationship needs were associated with high FCR and poor emotional and poor social function. Financial needs were associated with young age, high FCR, poor role function, and poor emotional function

Comment #32. In table 2, state what is in this column in a broad column heading (i.e. frequency (percentage)).

Response: Thank you for the comment. We modified Table 2 as below.

Table 2. Severity of Korean version of the Cancer Survivor’s Unmet Needs (CaSUN-K)

CaSUN-K’s survey items

Question Number

No need

Met need

Unmet need, weak

Unmet need, moderate

Unmet need, strong

Unmet need moderate-to-strong

Number (%)

Number (%)

Number (%)

Number (%)

Number (%)

Number (%)

Information

Up-to-date information

1

178 (18.8)

135 (14.2)

239 (25.2)

178 (18.8)

219 (23.1)

397 (41.8)

Information for others

2

221 (23.3)

108 (11.4)

256 (27.0)

177 (18.7)

187(19.7)

364 (38.4)

Understandable information

3

176 (18.5)

98 (10.3)

255 (26.9)

188 (19.8)

232 (24.4)

420 (44.3)

Comprehensive cancer care

Best medical care

4

238 (25.1)

154 (16.2)

175 (18.4)

172 (18.1)

210 (22.1)

382 (40.3)

Local health care services

5

228 (24.0)

124 (13.1)

175 (18.4)

192 (20.2)

230 (24.2)

422 (44.5)

Manage health with team

6

144 (15.2)

143 (15.1)

203 (21.4)

193 (20.3)

266 (28.0)

459 (48.4)

Communication among doctors

7

145 (15.3)

131 (13.8)

200 (21.1)

200 (21.1)

273 (28.8)

473 (49.8)

Complaints are addressed

8

236 (24.9)

127 (13.4)

176 (18.5)

185 (19.5)

225 (23.7)

410 (43.2)

Complimentary therapy*

9

405 (42.7)

80 (8.4)

199 (21.0)

134 (14.1)

131 (13.8)

265 (27.9)

Accessible hospital parking

18

557 (58.7)

95 (10.0)

95 (10.0)

66 (7.0)

136 (14.3)

202 (21.3)

Existential survivorship

  Reduce stress in my life

10

274 (28.9)

142 (15.0)

262 (27.6)

150 (15.8)

121 (12.8)

271 (28.6)

  Concerns about cancer recurrence

19

155 (16.3)

71 (7.5)

234 (24.7)

159 (16.8)

330 (34.8)

489 (51.5)

  Emotional support for me

20

352 (37.1)

102 (10.7)

218 (23.0)

166 (17.5)

111 (11.7)

277 (29.2)

  New relationships

23

470 (49.5)

107 (11.3)

184 (19.4)

102 (10.7)

86 (9.1)

188 (19.8)

  Talk to others

24

355 (37.4)

110 (11.6)

230 (24.2)

138 (14.5)

116 (12.2)

254 (26.8)

  Handle social/work situations

25

462 (48.7)

100 (10.5)

196 (20.7)

109 (11.5)

82 (8.6)

191 (20.1)

  Changes to my body

26

371 (39.1)

113 (11.9)

234 (24.7)

142 (15.0)

89 (9.4)

231 (24.3)

  Ongoing case manager*

28

490 (51.6)

68 (7.2)

167 (17.6)

110 (11.6)

114 (12.0)

224 (23.6)

  Move on with my life

29

424 (44.7)

124 (13.1)

216 (22.8)

105 (11.1)

80 (8.4)

185 (19.5)

  Changes to beliefs

30

422 (44.5)

111 (11.7)

228 (24.0)

114 (12.0)

74 (7.8)

188 (19.8)

  Acknowledging the impact

31

518 (54.6)

121 (12.8)

165 (17.4)

85 (9.0)

60 (6.3)

145 (15.3)

  Survivor expectations

32

415 (43.7)

140 (14.8)

209 (22.0)

101 (10.6)

84 (8.9)

185 (19.5)

  Decisions about my life

33

467 (49.2)

110 (11.6)

192 (20.2)

103 (10.9)

77 (8.1)

180 (19.0)

  Spiritual beliefs

34

546 (57.5)

136 (14.3)

134 (14.1)

82 (8.6)

51 (5.4)

133 (14.0)

  Make my life count

35

476 (50.2)

117 (12.3)

189 (19.9)

96 (10.1)

71 (7.5)

167 (17.6)

Quality of life

  Manage side effects

11

352 (37.1)

83 (8.7)

201 (21.2)

136 (14.3)

177 (18.7)

313 (33.0)

  Changes to quality of life

12

300 (31.6)

112 (11.8)

235 (24.8)

143 (15.1)

159 (16.8)

302 (31.8)

Relationships

  Support partner/family

21

410 (43.2)

100 (10.5)

203 (21.4)

133 (14.0)

103 (10.9)

236 (24.9)

  Impact on my relationship

22

450 (47.4)

92 (9.7)

188 (19.8)

122 (12.9)

97 (10.2)

219 (23.1)

  Changes to partner’s life*

13

900 (94.8)

23 (2.4)

11 (1.2)

6 (0.6)

9 (0.9)

15 (1.6)

  Problems with sex life

27

707 (74.5)

66 (7.0)

109 (11.5)

44 (4.6)

23 (2.4)

67 (7.1)

Financial issues (new dimension)

  Impact on my working life*

14

729 (76.8)

59 (6.2)

78 (8.2)

37 (3.9)

46 (4.8)

83 (8.7)

  Financial support*

15

502 (52.9)

77 (8.1)

141 (14.9)

104 (11.0)

125 (13.2)

229 (24.1)

  Life/travel insurance*

16

584 (61.5)

73 (7.7)

139 (14.6)

71 (7.5)

82 (8.6)

153 (16.1)

  Legal services*

17

600 (63.2)

53 (5.6)

118 (12.4)

92 (9.7)

86 (9.1)

178 (18.8)

*Seven items, originally not grouped into any dimension, were distributed into this new or an existing dimension.

Comment #33. In table 3, extend variables to independent variables. For each independent variable, put the reference category on a separate line above the non-reference category. The frequency (%) for each category can then be given. In terms of the OR and 95% CI, a 1.00 can be entered for all reference categories. For each domain, put aOR in the column and to the left add a column for OR. This is becaused the unadjusted OR from the univariable logiastic regression analysis are required. Please note that the unadjusted OR should be given for all variables in this table, not just the ones included in the multivariable model. For each domain, before the odds ratios and 95% CI, it would be best to add a column for the frequency (%) for each category of the given independent variables.

Response: Thank you for your insightful comment. Please refer to comment #20 and its mention of univariable analysis and modified Table 3.

Comment #34. State the type of study (e.g. cross-sectional study) here. Change predictors to correlates. Change lung cancer to NSCLC. State the exact percentage in words.

Response: Thank you for the comment. We updated the word as suggested.

[Discussion] (page12 , paragraph 1, line 191)

In the present cross-sectional study, the prevalence and correlates of unmet needs in NSCLC survivors were investigated. 91.8% of participants reported one or more unmet need.

Comment #35. In what population? Where was this study conducted? The word studies is used but only one reference (for a non-review study) is given. Please either: change studies to study, state (e.g.[13]), or,add one or more further references. Should this be in korea and other Asian countries? State in the where and elsewhere in this paragraph.

Response: Thank you for the comment. The given reference was conducted in Australia. The plural was changed to singular form. Our results represent a Korean sample, while the previous study was conducted across the Asia-Pacific region. We updated the sentences for clarity.

[Discussion] (page 12, paragraph 1, line 193)

The highest domains of unmet needs were existential survivorship (59.1%), comprehensive cancer care (51.2%), and information (49.7%), in agreement with the Australian population of the CaSUN.

[Discussion] (page 12, paragraph 2, line 202)

These results were consistent in the order of frequency with a previous study of patients across the Asia-Pacific region [14], in which the top five moderate-to-strong unmet needs in Korea were “concerns about cancer recurrence” in the existential survivorship domain and “local healthcare services,” “best medical care,” “manage health with team,” and “communication among doctors” in the comprehensive cancer care domain.

Comment #36. There is no corresponding reference in the reference list [26].

Response: Thank you for the comment. We updated the reference.

[Reference] (page 16, line 394)

  1. Clinical oncology network for unifying electronic medical data https://bigdata-cancer.kr/ncc/lungInfo.do.

Comment #37. Throughout this section, it is necessary to state whether all of the significant associations are supported by significant associations in past studies. If this is not the case for a particular association, then the significant association in the present study can be highlighted as a novel finding. How many of these studies [12,14, 27] looked at NSCLC specifically? How comparable are the past studies to the present study?

Response: None of these studies looked at NSCLC specifically. Nevertheless, in these previous studies “managing concerns about cancer recurrence” was the top unmet need in lung cancer survivors. Our finding is consistent with the previous results. “Concerns about cancer recurrence (51.5%) was the top moderate-to-strong unmet need in order of frequency.” Thank you for the insightful comments.

Comment #38. What are some examples of types of interactions and relevant health professions? What could be done in terms of models of care and polices?

Response: We appreciate your insightful comment. According to a systematic review and meta-analysis (Tauber et al., 2019), there were 25 previous interventions including 21 randomized controlled trials that evaluated the effects of psychological interventions on FCR among cancer survivors. Several interventions used traditional cognitive behavioral therapy (CBT), while other interventions included psychodynamic therapy or supportive therapy. These psychological interventions revealed a small but robust effect at post-intervention that was largely maintained at follow-up.

Another meta-analysis (Osborn et al., 2006) included not only CBT interventions, but also patient education such as information regarding the illness and symptom management and/or discussion of treatment options using booklets, video, or other educational materials. This study reported that individual interventions were more effective than group interventions.

Therefore, further optimized and tailored interventions such as psychological counseling or education could reduce the fear of cancer recurrence. Such programs can be enacted post-operatively for lung cancer patients.

[References]

Tauber NM, O’Toole MS, Dinkel A, et al. Effect of psychological Intervention on Fear of Cancer Recurrence: A systemic review and meta-anlysis. J Clin Oncol. 2019; 37(31):2899-2915.

Osborn RL, Demoncada AC, Feuerstein M. Psychological interventions for depression, anxiety, and quality of life in caner survivors: meta-analyses. Int J Psychiatry Med. 2006; 36(1):13-34.

[Clinical Implication] (page 13, paragraph 5, line 277)

Therefore, interventions of supportive care for NSCLC patients such as cognitive behavioral therapy or patient education should be focused on relieving FCR and improving emotional and social functioning.

Comment #39. The following limitations should also be highlighted: the possibility of social desirability and recall biases, give the use of self-reported measures. An issue in all observational studies. Any known confounding factors that were not adjusted for in this study should be named, with corresponding referencing.

Response: Thank you for your comments. We updated our limitations based on your suggestions.

[Limitation] (page 13, paragraph 6, line 284)

The present study had several limitations. First, the study cohort might not represent the general NSCLC survivor population because it included only patients from one tertiary hospital recruited using convenience sampling. In addition, patients who had previously experienced second cancer or cancer recurrence/metastasis were excluded. Therefore, a selection bias may limit generalization of these findings to other NSCLC survivors. Second, the study was conducted based on self-reported measures, so there could be social desirability and recall biases. Third, there is no consensus on the standard cut-off point of the EORTC-QLQ-C30 with cancer patients. A cut-off point <66.7 was used in the present study based on the population distribution of scores. In addition, this was an observational study and the cross-sectional settings restrict conclusion of any causal relationship between unmet needs and psychological traits such as FCR and poor functioning. Reciprocal relationships between unmet needs and QoL are also possible. Cohort studies are needed to elucidate such complex relationships. Finally, there was potential for unadjusted confounding factors

Comment #40. Give some examples of potential interventions.

Response: Refer to the answer of comment #39, that cognitive behavioral therapy (CBT), psychodynamic therapy, or supportive therapy are potential interventions (Tauber et al, 2019). Patient education such as information regarding the illness, symptoms, symptom management, and/or treatment options also are potential interventions (Osborn et al., 2006). Furthermore, some physical interventions such as yoga or dance could reduce the fear of cancer recurrence and improve emotional and social functioning. We appreciate your insightful comments.

[References]

Tauber NM, O’Toole MS, Dinkel A, et al. Effect of psychological Intervention on Fear of Cancer Recurrence: A systemic review and meta-anlysis. J Clin Oncol. 2019; 37(31):2899-2915.

Osborn RL, Demoncada AC, Feuerstein M. Psychological interventions for depression, anxiety, and quality of life in caner survivors: meta-analyses. Int J Psychiatry Med. 2006; 36(1):13-34.

Reviewer 3 Report

Thank you very much for inviting me to review the article titled “Unmet supportive care needs after non-small cell lung cancer resection”.

The article raises a very important issue of the needs of patients after oncological treatment. These issues are very rarely analyzed and discussed in the literature on the treatment of lung cancer.

The article is written in very good quality English, it may require only minor language corrections. The structure of the article is correct, it contains all necessary sections. In the introduction, the authors present the current state of knowledge and knowledge gaps, and present the objectives of the study. Selection of the study group, inclusion and exclusion criteria, questionnaires used and statistical analysis are correct. In the discussion, the authors carefully refer the obtained results to the current literature, analyze the significance of the results for the health of patients, outline further research directions and indicate the limitations of the study. I have only a few comments:

1. Please add an explanation of the “FCR” abbreviation in the footnotes of the Table 1. Although the abbreviation is explained in the table, I think it should also be described in the footnotes.

2. I don't quite understand the phrase: "The most frequently reported no need was...". Readers may also have trouble understanding the meaning of this sentence. Please clarify.

3. The quality of the Figure 1 is not sufficient, it is blurry. I would suggest to refine the image quality for the final text.

Once again, I would like to congratulate you on an interesting study addressing the issues of lung cancer treatment, which are rarely discussed in the literature and may be of great importance to patients.

Author Response

Response to comments from the Reviewer 3

Comment #1. Please add an explanation of the “FCR” abbreviation in the footnotes of the Table 1. Although the abbreviation is explained in the table, I think it should also be described in the footnotes.

Response: Thank you for the suggestion. We added footnotes of FCR in Table 1.

Comment #2. I don’t’ quite understand and phrase “The most frequently reported no need was…”. Readers may also have trouble understanding the meaning of this sentence. Please clarify.

Response: We deleted the sentences to avoid confusion. Thank you for the comment.

Comment #3. The quality of the Figure 1 is not sufficient, it is blurry. I would suggest to refine the image quality for the final text.

Response: We revised our image file of Figure 1 at the highest resolution of 300 dpi with no compression. We appreciate your thoughtful comment.
